# Expansive Soil Stabilization with Lime, Cement, and Silica Fume

**Ahmed S. A. Al-Gharbawi [1], Ahmed M. Najemalden [2] and Mohammed Y. Fattah [1,***

1   Civil Engineering Department, University of Technology-Iraq, Baghdad 00964, Iraq
2   Highways and Bridges Engineering Department, Technical College of Engineering, Duhok Polytechnic University, Duhok 00964, Iraq
*   Correspondence: 40011@uotechnology.edu.iq or myf_1968@yahoo.com

**Abstract:** The type of soil known as expansive soil is capable of changing its volume through swelling and contracting. These types of soils are mostly composed of montmorillonite, a mineral with the capacity to absorb water, which causes the soil to heave by increasing its volume. Due to their capacity to contract or expand in response to seasonal fluctuations in the water content, these expansive soils might prove to be a significant risk to engineering structures. Many studies have dealt with swelling soils and investigated the behavior of these soils, as well as their improvement. In this study, three percentages of lime, cement, and silica fume (5, 7, 9%) are used to stabilize the expansive soil, and the work is divided into two sections: the first is using a consolidation test to record the free swell and swell pressure for the untreated and treated soils; in the second part, the grouting technique is utilized as a process that can be applied in the field to maintain the improvement in the bearing capacity. It is concluded that the soil stabilized with different percentages of lime, cement, and silica fume exhibits a decrease in both free swell and swelling pressure by approximately 65% and 76%, respectively, as compared with untreated soil. The soil grouted with silica fume increases the bearing capacity of footings resting on the grouted soil by approximately 64% to 82% for the soil treated with 5% and 9% silica fume, respectively, as compared with untreated soil.

**Keywords:** expansive soil; lime; silica fume; stabilization; grouting

## 1. Introduction

Plate-shaped clay particles can be assembled in many ways. The ability of some clay particles to attract and hold water molecules on their surfaces and absorb them is of great interest. It is well known that water molecules exhibit a phenomenon known as polarization, in which each molecule has a distinct charge on its opposite sides, one positive and the other negative [1]. These polar water molecules adhere to each of the plate-shaped particles, forming a film of charged fluid on them. The swell or heave noticed in the swelling soil when the water content of this soil is increased is caused by the clay particles repelling one another as a result of the "double layer phenomenon", which emerges when distinct nearby particles are taken into account. Greater clay-particle-specific surfaces and higher charge densities make clay soils better suited to take water into their structure. The liquid limit and plastic limit indices are used to determine how well a cohesive soil can keep water molecules inside its structure. Montmorillonite clays have the greatest tendency to swell when compared to other clay minerals, whereas kaolinite has relatively low swelling potential [2].

She et al. [3] used a model test to study the swelling behavior of expansive soil at various elevations under fully saturated conditions of water injection from the top to the bottom. The results showed that there was a clear distinction in swelling between the expanding soil layers at various elevations during the saturation process and the electric charge effect, which comprises two variables, The primary causes of the swelling disparity between diverse expansive soil layers were (i) the transformation impact and

(ii) the electrical environment. The expansive soil's ability to expand may be lessened in the first scenario due to the cation exchange that occurs between expansive soil and aggregation bivalent cations. Additionally, the water film that has formed around the aggregates prevents additional water molecules from adhering into the montmorillonite interface, which halts the expansion of the swelling soil.

A swelling soil was treated by Fattah et al. [4], including a different range of additives, such as cement, steel fibers, gasoline and cement-based grout. Results were better when cement grout was injected or when the expansive soil was treated with 5% cement or steel fibers, although 4% gasoline oil is enough to show this material's best uses. The treatment does not affect the angle of internal friction, but the addition of these components caused a small variation in the adherence of the additive to the particles of the soil, which has a small impact on the cohesion between the particles.

A soil treated with waste fly ash was studied by Baloochi et al. [5]. The findings of this study, demonstrated that adding different percentages of waste fly ash causes stabilized soil to expand, but that the expansion can be slowed down by waiting for around 30 min before combining the material with water and compacting it.

Al-Soudany [6] used clayey soil mixed with 30, 50 and 70% of bentonite and stabilized the mixing soil with 3, 5 and 7% of nano-silica fume, and the findings showed that the Atterberg limits, specific gravity, maximum dry unit weight and optimum water content improved when increasing the nano-silica fume percentage.

When built on expansive soils, roads and other structures that are considered as light structures are significantly impacted. After a few years, the capacity of these soils to heave causes damage to these structures due to the swelling pressure, which causes cracking and swelling, with rapid lifting of the subgrade beneath the road and foundations causing cracks in the floor, walls and road that result from water seeping into the soil [7]. By utilizing the technique of stabilization of soil, which is a general term for any physical, chemical, or biological method, or any combination of these methods, used to enhance or change the specific characteristics of natural soil to make it usable for the intended engineering work, the risk posed by such soils can be reduced [8].

The moisture content of the soil changes as a result of precipitation or evapotranspiration in tandem with the climatic or seasonal changes in the region, known as the active zone or seasonal zone, which is sufficiently close to the ground surface. With the depth of the active zone, the major portion of the soil zone that experiences swelling phenomena increases [9]. The geography, climate, soil type and soil structure all have an impact on the depth of the active zone (depth of desiccation), which typically ranges between 1 and 4 m [10].

To analyze the features and behavior of this type of soil under circumstances that are comparable to those observed in the field, a variety of methodologies have been utilized to quantify the potential magnitude of swell in clay. Das [11] presented a simple laboratory test that is used to determine the magnitude of swelling pressure in soils, which is the "oedometer test". According to ASTM D 4546 [12], the sample is added to the oedometer cell with a modest surcharge of 6.9 kN/m$^2$, and water is then added to cause the soil sample to expand, which allows the volume to be measured until equilibrium is attained. It is possible to express the amount of free swell as a ratio:

$$S_{w(free)}(\%) = \frac{\Delta H}{H} \times 100 \qquad (1)$$

where:

$S_{W(free)}$: free swell as a percentage;
$\Delta H$: change in height of swell due to saturation;
$H$: original height of the specimen.

According to Negi et al. [13], if fine-grained clay soil passes through a 75 mm screen at least 25% of the time, has more than 0.3% sulfate, has a plasticity index above 10 and

contains more than 1% organic material, stabilization is required. Before it may be used as a sub-base, sub-grade or base for the construction of highways, bridges and many structures, expansive soil needs to be stabilized. The main goals of stabilizing the soil are to make the natural soil more rigid and hard and to lessen its flexibility and tendency for shrinkage and swelling [14]. According to Firoozi et al. [14], stabilized expansive clay soil has a higher bearing capacity when a heavy load is placed on it.

For expansive soils to have a lower likelihood of expanding, soil stability is essential [15]. Chemical stabilization seeks to offer additives that decrease and increase for both the liquid limit and plastic limit, respectively, as well as decreasing the plasticity index as a result of the liquid and plastic limits. As a result, the stabilized soil becomes more compressible, which improves the workability of the soil, the water content and the maximum dry density.

The shear strength of soils treated with cement is increased, but the liquid limit, plasticity index and swelling potential are all reduced [16]. Since only a tiny amount of cement is needed, stabilizing granular soils using cement has proven to be more effective and cost-effective. According to research, it is difficult to treat soils with a plastic index (PI) > 30 with cement; for this reason, lime is added before mixing to maintain the soil's workability. This study also demonstrated that when the cement concentration is increased from 0 to 12%, the unconfined compressive strength (UCS) improves, and the soil flexibility decreases, changing from 57.81% to 27.57%.

With the use of the F1 ionic stabilizer, the basic physical parameters and shear strength parameters of this reinforced expanding clay were examined from an engineering standpoint [17]. The expansive clay soil's water sensitivity, compaction properties and shear strength were all greatly enhanced by the F1 ionic soil stabilizer. The adequate water and F1 ionic soil stabilizer mixture was determined at $0.5 \text{ L/m}^3$. According to the findings, adding F1 ionic stabilized to expansive clay soil raised the plastic limit by around 46%, increased the maximum dry unit weight by around 6%, decreased the liquid limit by around 10% and increased both the ideal moisture content and plastic limit. In terms of cohesiveness and the angle of internal friction, the shear strength parameters were both raised by around 64 and 30%, respectively.

According to Liu et al. [18] who suggested a combined seepage–erosion water inrush model, a grouting thickness of 6 m is appropriate for various types of soils. These studies illustrate why the amount of water surge should be used as an evaluation metric for grouting effectiveness. For the grouting materials' slurry to permeate the soil, they must have a particular level of grout ability [19]. Technical problems including poor hole formation and challenging slurry diffusion must be addressed by the grouting procedure. For different flowing water circumstances that may arise during grouting, Liu et al. [18] recommended the selection criteria for injection materials and grouting volume per meter.

Wu et al. [20] studied the effect of grouting for a tunnel in a case study. The findings demonstrated that minimizing the amount of water surge is mostly achieved by the displacement of the support structure and the thickness of the curtain grouting. The tunnel's ability to stop water rapidly is essentially unaffected by a grouting thicknesses greater than 5 m.

The objective of this study is to stabilize an expansive soil with (5, 7 and 9%) lime, cement, and silica fume to reduce the free swelling and swell pressure. A grouting technique is used through a small-scale model to improve the bearing carrying capacity, as a technique that can be used in the field. The methodology adopted in this paper presents a practical method of utilizing cement, lime, and silica fume as grout to swell soil, so as to enhance its properties. Thus, the suggested method of applying a stabilizer through grouting can be considered as a new method.

## 2. Materials Used

### 2.1. Soil Characterization

The soil was obtained from a field located in the south of Baghdad city. The properties of the soil are shown in Table 1, and the distribution of grain size is illustrated in Figure 1. The ASTM specifications were followed for the determination of the soil properties.

**Table 1.** Soil properties.

| Property | Value |
|---|---|
| Natural water content ($w.c$%) | 5.0 |
| Liquid limit ($L.L$%) | 121 |
| Plastic limit (P.L%) | 26 |
| Plasticity index (P.I%) | 95 |
| Specific gravity (Gs) | 2.69 |
| Gravel (>4.75 mm)% | 0 |
| Sand (0.075 to 4.75 mm)% | 16 |
| Silt (0.005 to 0.075 mm)% | 34 |
| Clay (less than 0.005 mm)% | 50 |
| Max. dry unit weight (kN/m$^3$) | 17.5 |
| Optimum moisture content (%) | 16.5 |
| Soil symbols (USCS) * | CL |

* Unified Soil Classification System.

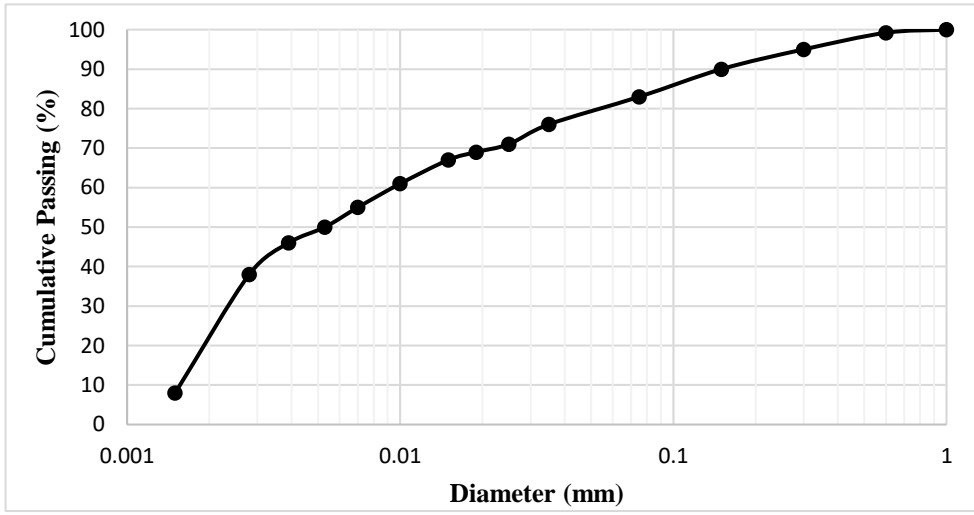

**Figure 1.** Grain size distribution of the natural soil.

*2.2. Lime*

Quicklime was used in this research and was obtained from Turkey. The properties of the lime are shown in Table 2.

**Table 2.** Lime properties.

| Index Property | Value |
|---|---|
| Physical properties | |
| Percentage of retaining in Sieve No. 30 | 0 |
| Percentage of retaining in Sieve No. 200 | 10 |
| Chemical properties | |
| CaO (%) | 93.35 |
| Free moisture content (%) | 0.09 |
| IR (%) | 2.0 |
| SO$_3$ (%) | 0.08 |
| LOI (%) | 25.25 |

### 2.3. Cement

Resistant Portland cement salt was used in this research. The properties of the cement are shown in Table 3. According to ASTM C 150, the cement was classified as type v.

**Table 3.** Cement properties.

| Index Property | Value |
|---|---|
| Strength after 7 days ($MN/m^2$) | 16 |
| Strength after 28 days ($MN/m^2$) | 28 |
| Initial setting (min.) | 92 |
| Final setting (min.) | 4.27 |
| $SiO_2$ (%) | 19.78 |
| CaO (%) | 63.7 |
| MgO (%) | 3.18 |
| $SO_3$ (%) | 2.16 |
| $C_3A$ (%) | 3.26 |
| LOI (%) | 0.88 |

### 2.4. Silica Fume

Silica fume was obtained from India. The properties of the silica fume as manufactured are shown in Table 4.

**Table 4.** Silica fume properties.

| Index Property | Value |
|---|---|
| $SiO_2$ (%) | 98.88 |
| $Al_2O_3$ (%) | 0.01 |
| $Fe_2O_3$ (%) | 0.02 |
| CaO (%) | 0.25 |
| MgO(%) | 0.01 |
| $K_2O$ (%) | 0.07 |
| $Na_2O$ (%) | 0.001 |
| Others (%) | 0.759 |

## 3. Testing Program

The work is divided into two parts as follows, and the testing program is shown in Figure 2.

The first group consisted of 10 tests and was used to investigate the free swell and swell pressure by using an oedometer test, as shown in Figure 3. The oedometer mold was prepared with the soil remolded at a unit weight of 17.5 $kN/m^3$ and water content of 16.5%, and then the mixing soil was stabilized with 5, 7 and 9% of lime, cement or silica fume. The percentages cement, lime and silica fume were recommended by many researchers [21–23].

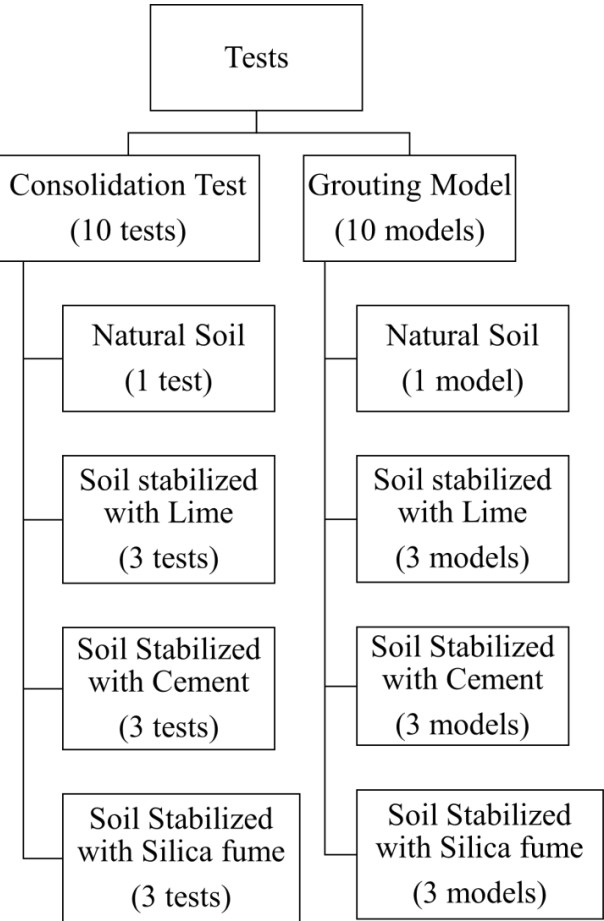

**Figure 2.** Testing program.

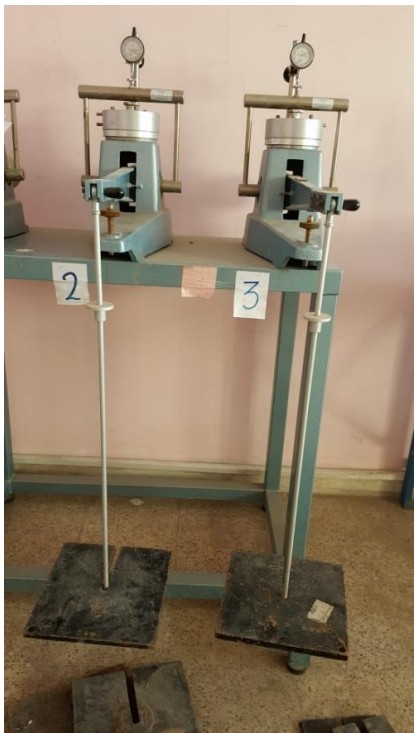

**Figure 3.** Consolidation test.

The other group consisted of 10 models and adopted the grouting process in the laboratory as a simulation of the field. The soil was prepared and mixed with bentonite and filled in a steel container of 400 × 400 × 400 mm dimensions. A footing of dimensions 60 × 60 × 10 mm was used to load the soil to investigate the load-carrying capacity of the soil and grouting soil with additives. A grouting probe of 10 mm in diameter and 120 mm in height was used to grout the liquid additives to the mixing soil. The grouting assembly is shown in Figure 4, and the grouting probe is shown in Figure 5. The mixing proportions are shown in Table 5. Each model was prepared as follows:

1.  The soil was divided into five parts in the steel container. The required percentage of lime, cement or silica fume was weighed and then mixed with water at a ratio of 1:2, as recommended by Al-Gharbawi et al. [24] (i.e., for 5% of the additive, we need around 1000 gm of the additive material weight and around 2000 mL of water; for 7% of additive, 1400 gm of additive material is required and 2800 mL of water, while, for 9% of additive, 1800 gm of additive material is mixed with 3600 mL of water).
2.  The slurry of additive material was poured into the slurry tank, which was closed completely.
3.  The compressor in the control panel was opened to control the pressure in the slurry tank, and the gauge reading was monitored to be between 0.15 and 0.25 bar.
4.  The grouting then started from the middle of the model to its edges to achieve the uniform distribution of the grouting slurry for all grouting holes.

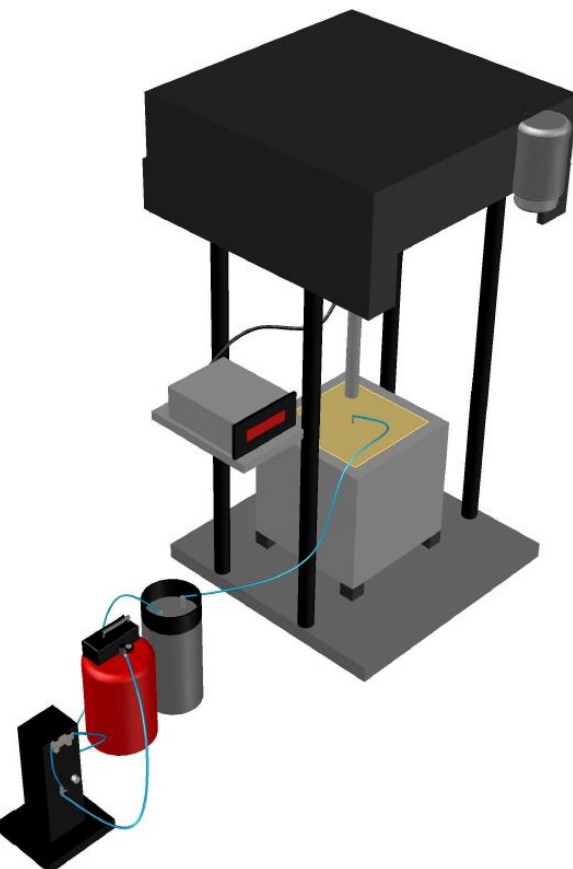

(**a**) Dimensional view

**Figure 4.** *Cont.*

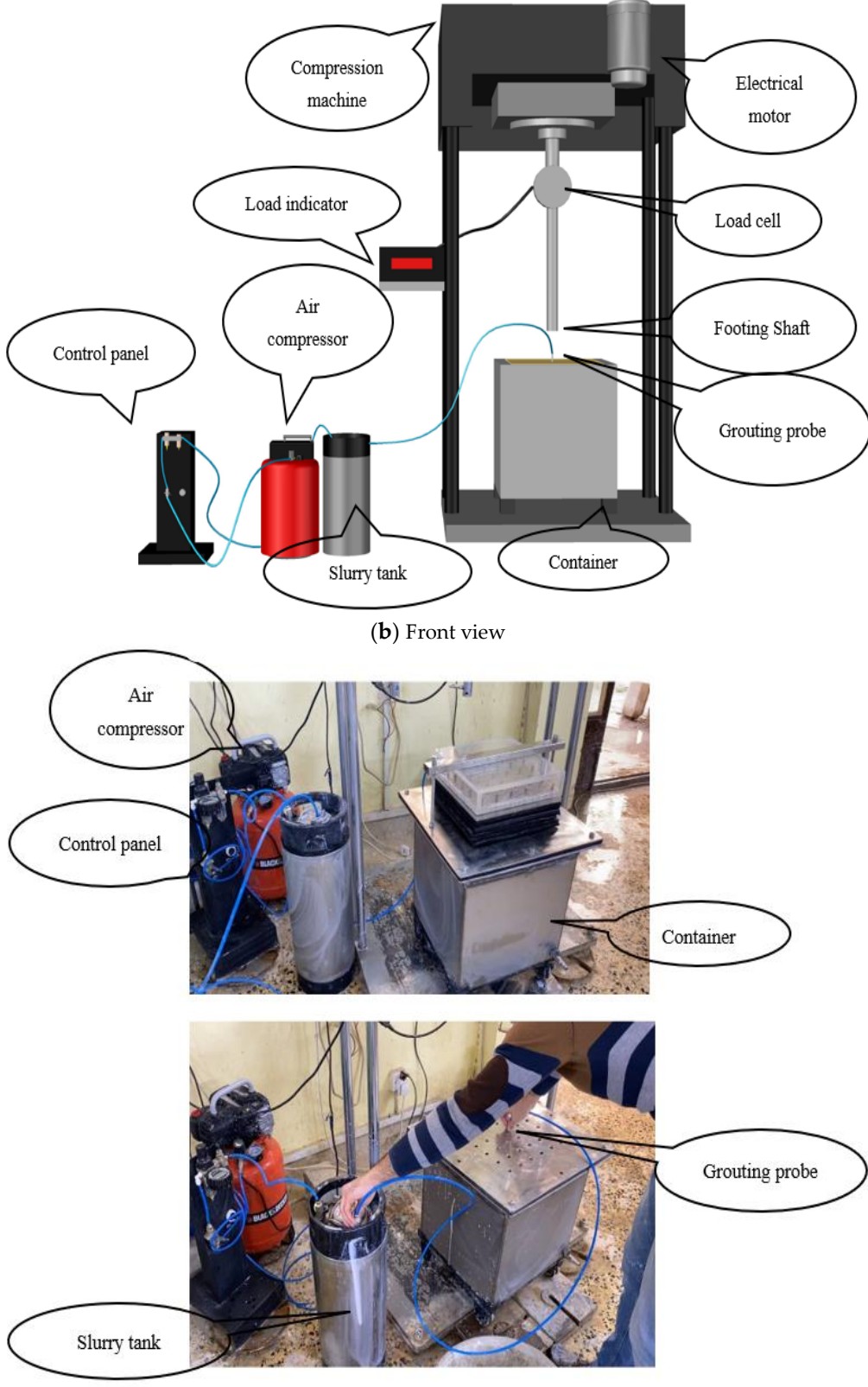

(**b**) Front view

(**c**) Real photos

**Figure 4.** Grouting assembly.

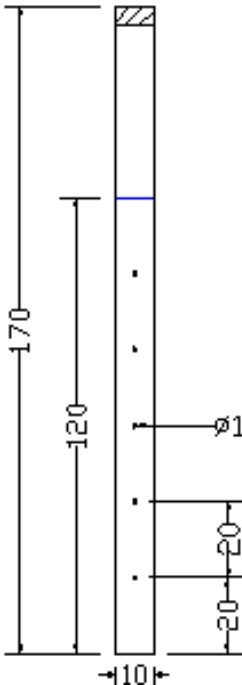

**Figure 5.** Grouting probe.

**Table 5.** Mixing proportions for 1 cubic meter of the grout material.

| Material | Density (kg/m$^3$) |
| --- | --- |
| Soil | 1750 |
| Lime | 3000 |
| Cement | 1400 |
| Silica fume | 500 |

## 4. Presentation and Discussion of Test Results

### 4.1. Consolidation Test

To measure the free swell and swelling pressure for the expansive soil and the expansive soil treated with lime, cement and silica fume, some samples were prepared and utilized as oedometer cells. In Figures 6 and 7, the free swell and swelling pressure for the untreated and treated soil are depicted. It is noticeable that when adding a stabilizing substance, such as lime, cement or silica fume, both the free swell and swelling pressure decrease. The rate of decrease seems to be uniform for the three types of additives.

With an increase in the percentage of additive, as seen in Figures 6 and 7, there is a decrease in both free swelling and swelling pressure. This results from an improvement in the chemical properties of the soil, which subsequently leads to an improvement in the overall soil and less swelling. It is caused by the chemical composition of the additive materials and the chemical characteristics of the expansive soil.

Due to the presence of Van der Waals forces joining the montmorillonite sheets, the soil expands as a result. When water is absorbed, this repulsion between the forces causes the soil volume to grow [25].

According to Ameta et al., the swelling pressure rises with a rise in the dry unit weight and falls with a rise in the initial water content of the molding (2008). The initial molding water content, particularly on the dry side of the optimum, has a greater impact on lowering or raising the swelling pressure than the dry unit weight does.



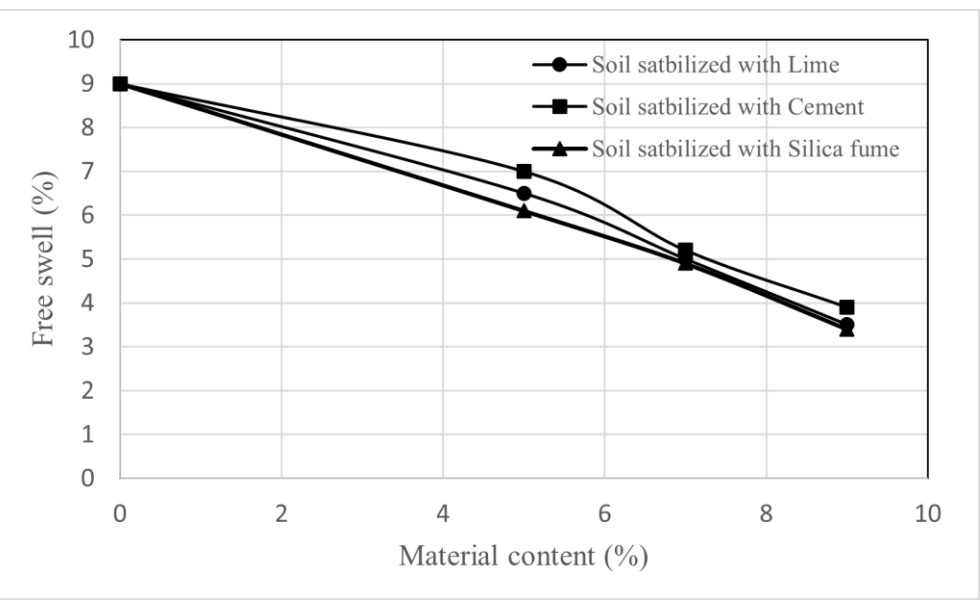

**Figure 6.** Free swell for different percentages of additives.

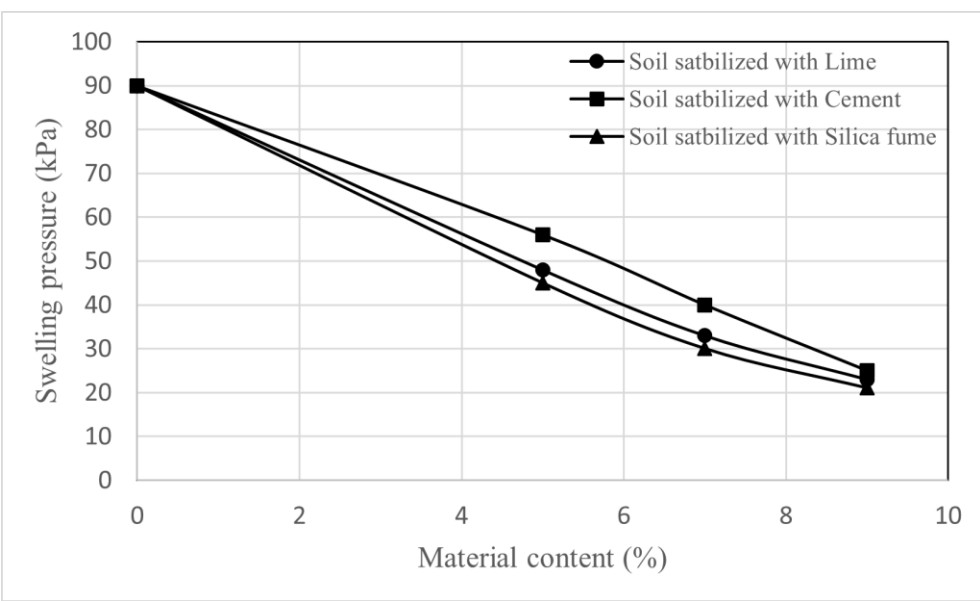

**Figure 7.** Swelling pressure for different percentages of additives.

The swelling pressure for materials compacted at high density (more than 1.4 Mg/m$^3$) can mostly be attributed to the pressures of hydration, as described by Kaufhold et al. [26].

The swelling percentage and swelling pressure have an inverse linear relationship with the initial water content when the initial dry unit weight is constant, as shown by Zumrawi [27] and Changiz et al. [28]. The swelling percentage and swelling pressure, on the other hand, may have a linear connection with the initial dry unit weight if the initial water content is constant.

### 4.2. Grouting Models

To simulate the field, it was suggested to maintain a process for grouting using lime, cement and silica fume. The selected percentage of materials was mixed with water and grouted in the soil at a pressure of 0.15–0.25 bar. The load–settlement relationship for untreated soil is shown in Figure 8. The load–settlement relationships for the soil treated with lime, cement and silica fume are shown in Figures 9–11. The failure load is considered

to correspond to a settlement of 10% of the footing width. The summary of the pressure at failure (i.e., at a 10% settlement ratio) is illustrated in Table 6.

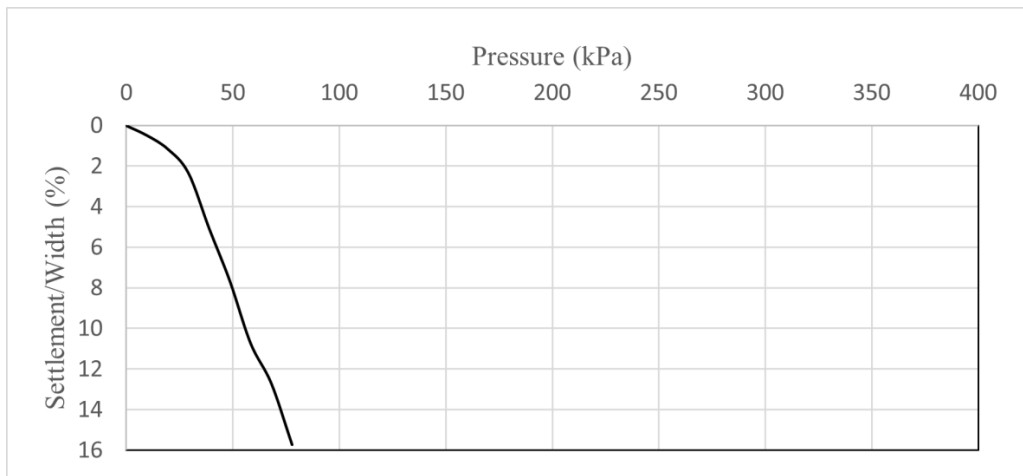

**Figure 8.** Pressure–settlement relationship for untreated soil.

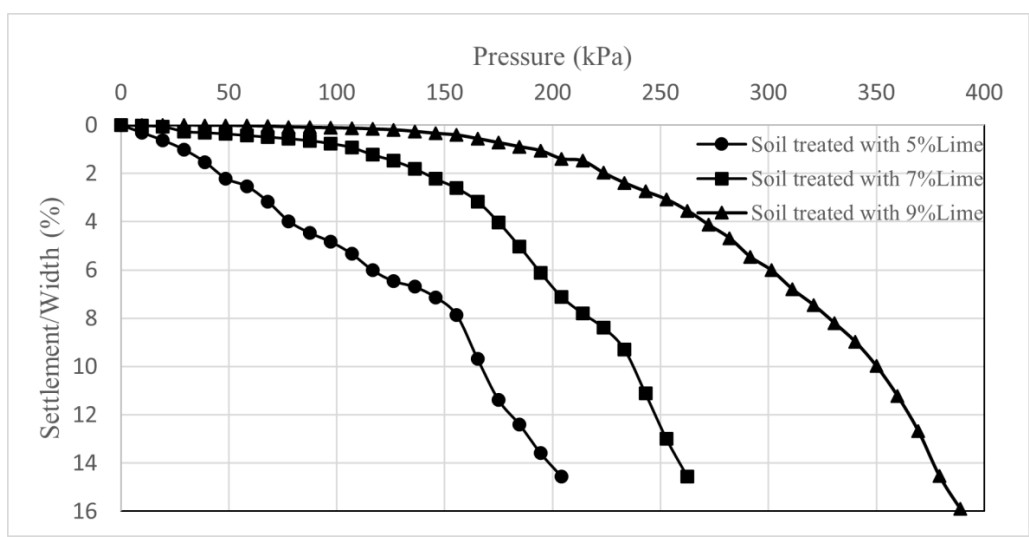

**Figure 9.** Pressure–settlement relationship for soil grouted with lime.

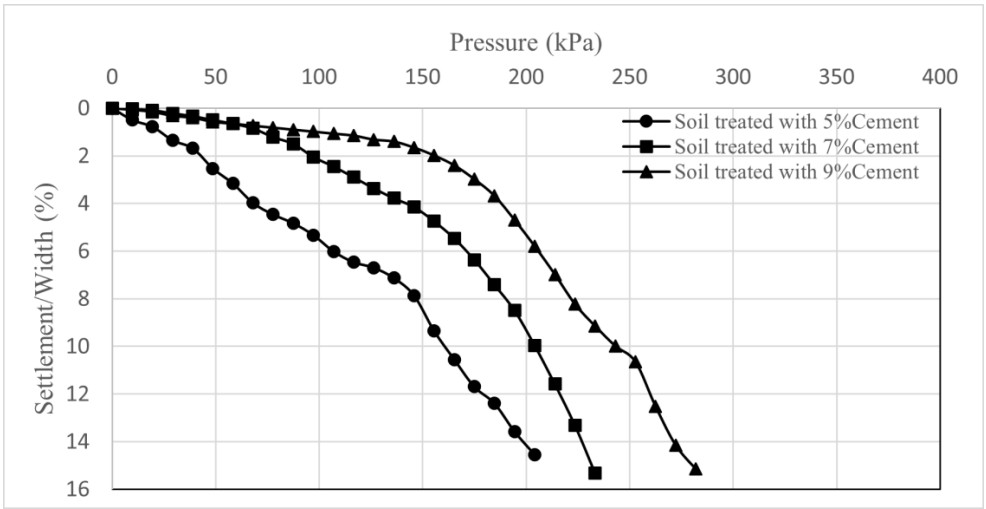

**Figure 10.** Pressure–settlement relationship for soil grouted with cement.

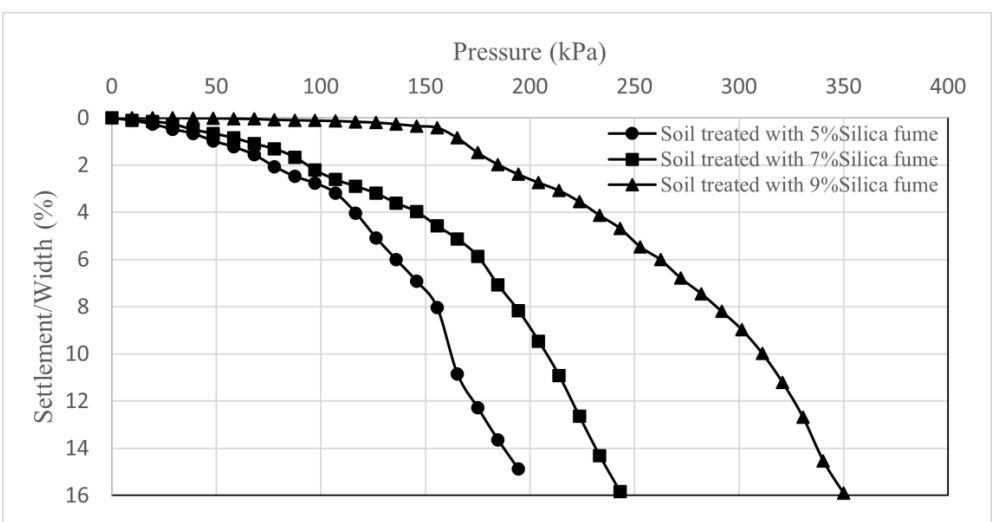

**Figure 11.** Pressure–settlement relationship for soil grouted with silica fume.

**Table 6.** Applied pressure at failure and bearing capacity ratio.

| | Pressure (kPa) | | Soil type |
|---|---|---|---|
| | 56.5 | | Untreated |
| 9% | 7% | 5% | Additive percent |
| 351.1 | 239.8 | 167.3 | Treated with lime |
| 249.4 | 205.3 | 160.9 | Treated with cement |
| 318.8 | 210.7 | 164.8 | Treated with silica fume |
| | Bearing capacity ratio (BCR) | | Soil type |
| 6.203 | 4.244 | 2.956 | Treated with lime |
| 4.406 | 3.634 | 2.843 | Treated with cement |
| 5.632 | 3.729 | 2.912 | Treated with silica fume |

The bearing capacity ratio is defined as follows:

$$BCR = \frac{Bearing\ capacity\ of\ footing\ on\ treated\ soil}{bearing\ capacity\ of\ footing\ on\ untreated\ soil} \tag{2}$$

The pozzolanic reactions that occur in the lime–soil, cement–soil or silica fume–soil mixtures that lead to strength gain over time are the causes of the reduction in swelling (the liberated silica and alumina combine with the calcium from the lime to produce cement) [22].

Overall, the lime, cement or silica fume stabilization process's flocculation and agglomeration phases produce soil that is easier to mix, work with and finally compact.

This lime is believed to be helpful as a soil stabilizer, a mortar binder, a neutralizing agent for water and sewage treatment and a method to maintain alkaline conditions while processing minerals. A well-known stabilizing ingredient is $Ca(OH)_2$, which is hydrated high-calcium lime that is readily available locally. When it comes to soil stabilization, hydrated lime has been widely shown to be superior to other types of lime. This is due to its straightforward application and small particle size, which facilitates soil blending.

The behavior of grout could influence the strength of the entirety [29,30], which could be reviewed with relative studies. Uncertainty analysis [31] and complex stress conditions [32] are important challenges to geotechnical analysis. The results could be applied in the future with optimum parameters in a field study considering unknown stress and environmental conditions.

In the work of Yue et al. [33], an expansive soil was stabilized using additives such as cement, zeolite powder, steel slag, fly ash, and blast furnace slag, and its affects on the swelling potential were examined at various curing durations and additive concentrations. The direct shear test was used to examine how the additives affected the strength of the swelling soil. The findings showed that the specimens' no-loading swelling potential could be greatly reduced by 82.5% over the course of 28 days of curing, and their cohesiveness could be significantly boosted by 82% with a 9% cement concentration. Cement can also change clay minerals that are hydrophilic into clay minerals that are non-hydrophilic. When a steel slug was used, the swelling potential was reduced. Ions had to be absorbed to diminish the adsorption of water molecules to the surface of the clay slices.

With many interconnecting pore spaces between soil particles inside the clods of smaller particles, fine-grained soil, such as silt or clay, that has been compacted at the appropriate dryness, often has an open structure. Large suction values are offered by these pores' size. However, specimens of wetness with optimum initial water content have no visible inter-clod pores and offer little suction, and they have an occluded structure. The measurements of the maximum amount of water that can be absorbed or desorbed by capillary action increase with the increase in the pore size [34].

On the other hand, Fattah et al. [35] concluded that silica fume at all percentages could increase the sand and silt particle sizes due to the pozzolanic reaction and large silica-fume-densified particles.

## 5. Conclusions

The objective of this study was to stabilize an expansive soil with lime, cement and silica fume, to reduce the free swelling and swell pressure. The methodology adopted in this paper presents a practical method of utilizing cement, lime and silica fume as grout to swell soil in order to enhance its properties. Thus, the suggested method of applying the stabilizer through grouting can be considered as a new method.

Three percentages of lime, cement and silica fume (5, 7, 9%) were used to stabilize the expansive soil. The work was divided into two stages: the first used a consolidation test to record the free swell and swell pressure for the natural and stabilized soils; in the second part, the grouting technique was utilized as a process that can be used in the field to maintain the improvement in the bearing capacity. From the test results, the following can be concluded.

1. From the laboratory swelling test, the free swelling and swelling pressure are increased rapidly for untreated soil.
2. The soil treated with lime, cement and silica fume had a reduction in both free swell and swelling pressure of around 65 and 76%, respectively, as compared with untreated soil.
3. According to grouting models, when a footing is supported by soil that has been grouted with lime, the bearing capacity of the soil increases from 66 to 85% when compared to untreated soil for soil that has been treated with 5 and 9% lime, respectively.
4. The soil grouted with cement increased the bearing capacity of footings by around 60 to 75% for the soil treated with 5 and 9% cement, respectively, as compared with untreated soil.
5. The soil grouted with silica fume increased the bearing capacity of footings by around 64 to 82% for the soil treated with 5 and 9% silica fume, respectively, as compared with untreated soil.

## 6. Recommendations

The grouting technique is one of the most promising techniques to improve the properties of soil. The grouting technique is adopted in this study to improve the strength of expansive soil by using different percentages of lime to improve the soil's shear strength, reduce settlement and reduce both the free swelling and swelling pressure.

**Author Contributions:** Conceptualization, M.Y.F.; methodology, A.S.A.A.-G. and M.Y.F.; validation, A.S.A.A.-G. and M.Y.F.; formal analysis, M.Y.F.; investigation, A.M.N.; resources, A.M.N.; data curation, A.M.N.; writing—original draft preparation, A.M.N.; writing—review and editing, A.S.A.A.-G. and M.Y.F.; visualization, A.S.A.A.-G.; supervision, M.Y.F.; project administration, A.S.A.A.-G.; funding acquisition, A.M.N. All authors have read and agreed to the published version of the manuscript.

**Funding:** This research received no external funding.

**Institutional Review Board Statement:** Not applicable.

**Informed Consent Statement:** Not applicable.

**Data Availability Statement:** The data that have been used are confidential.

**Conflicts of Interest:** The authors declare no conflict of interest.

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
