# Peer review of "Expansive Soil Stabilization with Lime, Cement, and Silica Fume"

_applsci, doi:10.3390/app13010436_

Round 1

Author Response

Response to reviewers comments is listed in attached file.

Reviewer 2 Report

The paper is not readable from perspectives of English editing, structures. The results are not well presented and in lack of laboratory photos. The novelty is not so strong. The comments are listed as follows. 

(1) The background of the study needs to be improved. What is the lack in the preview and the novelty of this study. Why should the study be conducted?

(2) English editing needs improvement, especially in the abstract as the readers may read it first.

(3) The used laboratory materials should be shown. The preparation of materials should be expressed with figures.

(4) The laboratory test should be presented with photos, not just only a schematic diagram.

(5) How are the soils grouted? What is the process and corresponding photos?

(6) Although the results contain data, the results seem to be expected even without the study.

Author Response

(The authors gave the same response as above.)

Round 2

Author Response

(The authors gave the same response as above.)

Reviewer 2 Report

Please see the following comments to make the manuscript more suitable for publication before acceptance.

(1) Please list in details the items in Figure 2, e.g. what are the 10 tests as well as 10 models, the variation in each test and model. The authors should make it more clear that even without the looking for the text the test schemes could be understood.

(2) The authors are suggested to specify the contents or weight of the specimens, such as the mixing soil stabilized with 5, 7, and 9% of lime, cement, or silica fume. The contents could be listed in a table.

(3) How to make it practical from test to field application? For example, in the test the ratio of soils and addictives is constant. For the field, how to calculate and make sure exactly the weight of soil and the addictive ratio?

(4) The behaviour of grout could influence the strength of entirety (Study of grouting effectiveness based on shear strength evaluation with experimental and numerical approaches; Grouting effect on rock fracture using shear and seepage assessment), which could be reviewed with the relative studies. Uncertainty analysis (A practical and efficient reliability-based design optimization method for rock tunnel support) and complex stress conditions (Unloading behaviors of shale under the effects of water through experimental and numerical approaches) are important challenges to geotechnical analysis . The results could be applied in future with optimum parameters with field study considering the unknown stress and environmental conditions. The discussions could be added.

(5) Figure 4 is suggested to be described with the parts listed in Figure 4(b) in details.

(6) The description of Figure 5 is suggested to be added combined with Figure 4.

Other editorial errors:

(1) Line 202: "was" should be deleted.

(2) Line 216: Figure 3 should be written as figure 4.

Author Response

(The authors gave the same response as above.)

Round 3

Reviewer 1 Report

The authors properly addressed my comments and significantly improved the manuscript quality. Although I have no further technical comment, I suggest to the authors to proofreading their paper one more time.

Reviewer 2 Report

I appreciate the authors for trying to address my questions. I feel the quality has been improved.